# Primary HIV-1 and Infectious Molecular Clones Are Differentially Susceptible to Broadly Neutralizing Antibodies

**DOI:** 10.3390/vaccines8040782

**Published:** 2020-12-21

**Authors:** Jiae Kim, Venigalla B. Rao, Mangala Rao

**Affiliations:** 1US Military HIV Research Program, Henry M. Jackson Foundation for the Advancement of Military Medicine, 6720A Rockledge Drive, Bethesda, MD 20817, USA; jakim@hivresearch.org; 2Laboratory of Adjuvant and Antigen Research, US Military HIV Research Program, Walter Reed Army Institute of Research, Silver Spring, MD 20910, USA; 3Department of Biology, The Catholic University of America, 620 Michigan Avenue NE, Washington, DC 20064, USA

**Keywords:** HIV-1, virus capture, broadly neutralizing antibodies, primary viruses, infectious molecular clones, PBMCs, qRT-PCR

## Abstract

To prevent the spread of HIV-1, a vaccine should elicit antibodies that block viral entry for all cell types. Recently, we have developed a virus capture assay to quantitatively examine early time points of infection. Here we present data on the ability of bNAbs to inhibit capture (1 h) or replication (48 h) of purified primary acute or chronic HIV or infectious molecular clones (IMCs) in human peripheral blood mononuclear cells (PBMCs) as quantified by qRT-PCR. Although bNAbs significantly inhibited HIV-1 replication in PBMCs in a virus subtype and in a PBMC-donor specific manner, they did not inhibit virus capture of primary viruses. In contrast, IMC capture and replication in PBMCs and purified CD4^+^ T cells were significantly inhibited by bNAbs, thus indicating that unlike IMCs, primary HIV-1 may initially bind to other cell surface molecules, which leads to virus capture even in the presence of bNAbs. Our results demonstrate that the initial interactions and some aspects of infectivity of primary HIV-1 and IMCs are inherently different, which underscores the importance of studying virus transmission using primary viruses in *in vitro* studies, an issue that could impact HIV-1 vaccine design strategies.

## 1. Introduction

Vaccines are the most effective way to prevent infectious diseases. Despite decades of work in the human immunodeficiency virus (HIV) field and several Phase 2b/3 vaccine clinical trials, the quest for an effective HIV-1 vaccine continues. A vast amount of data have been generated on HIV-1 envelope (env) glycoprotein and its interaction with broadly neutralizing antibodies (bNAbs). These bNAbs have been shown to bind to various regions of HIV-1 env and neutralize the virus in *in vitro* studies using the TZM-bl neutralization assay [1]. The utilization of pseudovirus or infectious molecular clones (IMC) in the neutralization assay demonstrates that bNAbs inhibit the initial interactions between HIV-1 env and the CD4 receptor on the host cell [2,3,4,5,6,7,8,9,10]. However, it should be noted that the *in vitro* cell line-produced pseudoviruses and IMCs, which are used for assessing neutralization, are probably significantly different from the quasispecies of HIV-1 that humans are exposed to during an infection. Furthermore, the read out for the TZM-bl neutralization assays is days post infection [11], far removed from the initial interactions, which probably occurs on the order of minutes in a natural infection in humans.

To address some of these gaps, we have developed a quantitative, kinetically controlled, and high throughput qRT-PCR assay that quantifies the number of HIV-1 genomes that enter human T lymphoblastoid (A3R5.7) cells at very early time points of infection [12]. This assay showed that the initial interactions between the virus and the host cell occurred rapidly, within minutes of exposure, and demonstrated differences in the ability of bNAbs to inhibit virus capture depending on whether it was a primary virus, a pseudovirus, or an IMC. In order to validate the innate differences between primary viruses and IMCs and that their interaction with the host cell was not restricted to a T cell line, but was physiologically relevant, we examined the initial interactions between the virus and human peripheral blood mononuclear cells (PBMCs) using primary HIV-1 and IMCs. Our results indicate that capture and/or replication of primary HIV-1 and IMCs in PBMCs are differentially affected by bNAbs.

## 2. Materials and Methods

### 2.1. Ethics Statement

RV229B (WRAIR Protocol #1386): This protocol “Apheresis of blood components from healthy volunteers for *in vitro* research” and all related documents were approved by the following independent Institutional Review Boards (IRBs): The Division of Human Subject Protection, Walter Reed Army Institute of Research, and the Ethical Review Committee for Research in Human Subjects. All volunteers provided written informed consent following discussion and counseling by the clinical study team prior to enrollment and before any blood draws were performed. The investigators have adhered to the policies for protection of human subjects as prescribed in AR 70–25.

### 2.2. Monoclonal Antibodies

The following reagents were obtained through the NIH AIDS Reagent Program, Division of AIDS, NIAID, NIH: Anti-HIV-1 gp120 monoclonal (PG9) from Dr. Hermann Katinger, and Anti-Human CD3/8 Bi-specific monoclonal from Drs. Johnson Wong and Galit Alter. The monoclonal antibody VRC01 was a gift from Dr. John Mascola, VRC, NIH. Synagis was purchased from MedImmune.

### 2.3. Media and Cell Culture

Media components and reagents were obtained as follows: RPMI-1640, L-glutamine, penicillin/streptomycin (Quality Biologicals Inc., Rockville, MD, USA), fetal bovine serum (Gemini Bio Products, CA, Carlsbad, USA), and recombinant human IL-2 (Roche/Sigma-Aldrich, Bethesda, MO, USA). PBMCs were stimulated for proliferation using a combination of the bi-specific anti-CD3/CD8 monoclonal antibody (final 0.3 µg/mL) and recombinant human IL-2. PBMCs were grown in RPMI-1640 containing 10% FBS, 1% L-Glutamine, 1% Penicillin/Streptomycin, and 50 U/mL IL-2 at 37 °C and 5% CO_2_. The cells were thawed, stimulated, and incubated for 24 h in 10 mL of media. Another 10 mL of IL-2-containing media was added to the cells and incubated for 72 h. Four days after the cells had been thawed, the stimulated PBMCs were used for the virus capture and replication assay.

### 2.4. HIV-1 Preparation and Purification

PBMCs were isolated by Ficoll density gradient centrifugation from healthy HIV-1 seronegative donors under the IRB-approved protocol, RV229/WRAIR number 1386. HIV-1 stocks of primary isolates BaL (subtype B), 40100 (subtype CRF01_AE), 40353 (subtype B), and TZBD9/11 (subtype C) were grown in CD3/CD8 antibody stimulated PBMCs. The BaL primary virus stock was obtained through the NIH AIDS Reagent Program, Division of AIDS, NIAID, NIH from Dr. Suzanne Gartner, Dr. Mikulas Popovic and Dr. Robert Gallo. The 40100 and 40353 primary virus stock was obtained from Dr. Victoria Polonis, USMHRP, WRAIR. The TZBD9/11 primary virus stock was obtained through the NIH AIDS Reagent Program, Division of AIDS, NIAID, NIH: HIV-1 TZBD9/11 (Cat# 11257) from Dr. Victoria Polonis. Primary HIV-1 BaL and Primary HIV-1 TZBD9/11 was harvested on day 4 post-infection, and 40100 and 40353 were harvested on day 14 post-infection. After the virus was harvested, it was purified, and stored as previously described [13]. The amount of viral p24 was determined using the HIV-1 p24 Antigen Capture Assay kit (ABL, Rockville, MD, USA). The infectivity of the virus was determined using the P4R5 MAGI assay [14]. The BaL IMC (subtype B) plasmid was obtained through the NIH AIDS Reagent Program, Division of AIDS, NIAID, NIH from Dr. Bryan R. Cullen and expanded by Genscript (Piscataway, NJ, USA). The plasmid was transfected in 293T cells using Fugene6 (Promega, Madison, WI, USA), harvested after 48–72 h of incubation, and then purified as previously described [13]. The C6980V0C22 (subtype C) and 40061.LucR (subtype CRF01_AE) IMCs were obtained from Dr. Sodsai Tovanabutra from USMHRP, Henry M. Jackson Foundation, and were also purified as previously described [13].

### 2.5. HIV-1 Capture Assay Using PBMCs

CD3/CD8 stimulated PBMCs grown in IL-2 RPMI media were harvested and then plated onto a 96 well polypropylene plate at 1 million cells (100 µL) per well. Purified virus and media (up to 50 µL) were added to the wells and the plates were incubated at 37 °C/5% CO_2_ for 1 h. In the experiments where bNAb inhibition was examined, the bNAb was added to the virus and media, and pre-incubated at 37 °C for 15 min. The amount of primary virus used for the capture assay ranged from 0.1 ng to 0.5 ng (p24 concentration) and the amount of IMC used for the capture assay ranged from 5 ng to 40 ng (p24 concentration). The amount of virus used for the capture assay was determined by the presence of ~ 50,000 viral RNA copies present per million cells after the 1-h incubation and trypsin treatment. The final concentrations used for each of the antibodies were: PG9 (5 µg/mL), VRC01 (10 µg/mL), and Synagis (5 µg/mL). After the cells were incubated with the virus, they were centrifuged at 1200 rpm for 5 min at 4 °C. The supernatant containing the virus was removed and the cells were washed, and trypsin treated as previously described [12]. Briefly, the cells were washed three times with ice-cold PBS, and then incubated with 0.05% trypsin at room temperature for 5 min. The treated cells were washed with media containing 15% FBS and then with ice-cold PBS. The resulting cell pellets were frozen at −20 °C, and later processed for RNA extraction. All experiments were performed with triplicate samples at least twice for each donor PBMCs.

### 2.6. HIV-1 Replication Experiments

Virus replication assay was performed with modifications to the virus capture assay described above. After washing the cells following trypsin treatment, cells were resuspended in a total volume of 1 mL of IL-2 media either lacking or containing the bNAbs at the same final concentration as noted above for the capture assay. The cells were transferred to a 24 well plate and incubated at 37 °C/5% CO_2_ for 48 h.

### 2.7. Isolation of CD4^+^ T Cells

Frozen PBMCs were thawed, stimulated with CD3/CD8 bispecific antibody and grown for four days in IL-2 media at 37 °C/5% CO_2._ CD4^+^ T cells were isolated from bulk PBMCs using the Easy Sep Human CD4 T cell magnetic isolation kit (StemCell Technologies, Vancouver, BC, Canada). The negatively selected CD4^+^ T cells exhibited ~95% purity as determined by flow cytometry. Purified CD4^+^ T cells were then plated for the virus capture assay and replication experiments as described above.

### 2.8. RNA Extraction and qRT-PCR

RNA was extracted from the infected cell pellets using the RNeasy Mini Kit (Qiagen, Hilden, Germany). The RNA samples were eluted using RNase Free water and analyzed using the Nanodrop (Thermo Fisher, Waltham, MA, USA) for nucleic acid concentration and 260/280 nm ratio. All samples showed a 260/280 ratio of greater than 1.8. The samples were stored at −30 °C. The two-step real-time RT-PCR assay was performed with a Viia7 (Applied Biosystems, Foster City, CA, USA) using the Taqman RNA-to-Ct kit (Applied Biosystems). The reaction mixture (50 µL) contained the following amounts of reagents: 200 ng of total RNA, 1X final concentration of the Taqman RT-PCR Mix and Taqman RT Enzyme Mix, 0.2 µM LTR forward primer, 0.2 µM LTR reverse primer, 0.2 µM LTR probe, 0.2 µM GAPDH forward primer, 0.2 µM GAPDH reverse primer, 0.2 µM GAPDH probe. The reactions were run using the following program: 48 °C for 20 min, 95 °C for 10 min, (45 cycles of) 95 °C for 15 s, 59 °C for 1 min. The primer-probe set used for HIV-1 quantification was: LTR forward: 5′-GCCTCAATAAAGCTTGCCTTGA-3′, LTR reverse: 5′-GGGCGCCACTGCTAGAGA-3′, LTR probe: 5′-FAM-CCAGAGTCACACAACAGACGGGCACA-BHQ-3′. GAPDH forward: 5′-GAAGGTGAAGGTCGGAGTCAAC-3′, GAPDH reverse: 5′-CAGAGTTAAAAGCAGCCCTGGT-3′, GAPDH probe: 5′-HEX-TTTGGTCGTATTGGGCGCCT-BHQ-3′. Primers and probes were purchased from Integrated DNA Technologies, IA, USA.

For each run, two standard curves, one using the diluted HIV-1 RNA transcripts and the other using the diluted cellular RNA were generated. The viral RNA and cellular RNA standards were prepared as previously described [12]. The threshold cycle (Ct) values were plotted as a function of the input transcript copy number, and linear regression was plotted. Controls lacking RNA template were run to test contamination with the PCR product during the assay setup. The numbers of copies of HIV-1 and cells were calculated by interpolation of the experimentally determined Ct value by using the standard curves. For HIV-1 quantification, the calculated number of copy equivalents per reaction was determined. For cell number, the calculated number of cells per reaction was determined. The number of HIV-1 copies was normalized to the number of cells present in the reaction. The final value was expressed as number of copies per million cells. Assay acceptability was contingent on the R^2^ value for the HIV-1 and cell linear regressions were >0.95.

The percentage inhibition by the bNAb or Synagis was determined by comparing the viral RNA copies associated with the cells in the presence of the antibody to that in their absence: (100-(viral RNA copies per million cells in the presence of antibody/viral RNA copies per million cells in the absence of antibody) * 100)).

### 2.9. Statistical Analysis

The data in the figures was graphed and analyzed using GraphPad Prism, version 8.0 (GraphPad Software). The percentage of inhibition was plotted as either box-whiskers and showing all data points or as bar graphs represented as mean ± SEM. Reported percentage inhibition values are the median values in the box whisker plots or mean ± SEM in the bar graphs. An unpaired t-test was utilized to determine statistical significance comparing bNAb inhibition to inhibition observed with the negative control antibody, Synagis.

## 3. Results

### 3.1. Capture and Replication of Primary HIV-1 in Human PBMCs

Human PBMCs were obtained from healthy HIV-1 seronegative donors and were stimulated using the CD3/CD8 bispecific antibody. The cells were cultured in IL-2 media for 4 days before HIV-1 exposure. The virus capture assay was performed as shown in Figure 1 and as previously described [12]. Briefly, the stimulated PBMCs were incubated with purified virus [13], washed, and then trypsin treated at room temperature. The trypsin-treated cells were washed and processed for RNA extraction and the viral RNA copies were determined using qRT-PCR (virus capture). Alternatively, the cells were incubated for 48 h, washed, and then processed for RNA extraction (replication) as above.

Using this assay, we examined the ability of the V1V2-specific bNAb, PG9, and the CD4-binding site bNAb, VRC01, to inhibit virus capture and replication in PBMCs from three different donors. Anti-Respiratory Syncytia Virus (RSV) antibody, Synagis, served as the negative control mAb in the experiments. Similar to what we had previously reported with A3R5.7 cells [12], the two bNAbs were unable to inhibit the capture of primary BaL (subtype B) on the PBMCs (Figure 2A). Although we observed slight differences due to donor-to-donor variation, we did not observe inhibition of more than 22% with either PG9 or VRC01. However, the two bNAbs inhibited primary BaL virus replication after 48 h of infection (Figure 2B) by over 80%. We also utilized a chronic subtype C virus, TZBD9/11, to examine virus capture and replication. Both PG9 and VRC01 were unable to inhibit capture of primary HIV-1 TZBD9/11 on the PBMCs (Figure 2C). However, the two bNAbs inhibited primary TZBD9/11 virus replication after 48 h of infection (Figure 2D) by over 90%.

To confirm that the inability of bNAbs to inhibit primary HIV-1 BaL (a chronic virus) capture was not due to the time of isolation based on the infection status (chronic), we examined the ability of bNAbs to inhibit virus capture of primary acute HIV-1. We utilized primary acute HIV-1 from the USMHRP RV217 Early Capture HIV Cohort (ECHO) study [15]. This study followed high-risk volunteers and tracked their HIV status, which enabled the ability to identify viruses at the earliest stages of HIV-1 infection. Two acute viruses, 40100 (subtype CRF01_AE) and 40353 (subtype B), from this study were utilized in the capture assay. Utilization of these two viruses also enabled us to determine if acute primary viruses behaved differently from BaL. We examined the ability of the bNAbs to inhibit virus capture and replication with bulk PBMCs. The capture of primary acute subtype B virus 40353 by PBMCs, showed none (PG9) to very minimal inhibition (14% with VRC01) with donor 002 (Figure 2E). VRC01 was more effective than PG9 at inhibiting virus replication of the subtype B 40353 for all three donor PBMCs (56–72%). PG9 inhibited virus replication in two of the donor PBMCs (24%), while it was ineffective in PBMCs of donor 002 (Figure 2F). Similarly, no inhibition of virus capture by PBMCs was observed with acute primary HIV-1, 40100 with either PG9 or VRC01 (Figure 2G). In contrast to virus capture, both PG9 and VRC01 inhibited virus replication, albeit to varying degrees. PG9 inhibited replication of the CRF01_AE virus 40100 for all three donor PBMCs (48–75%), while VRC01 inhibited replication in one out of three donor PBMCs (63%) (Figure 2H). These results further demonstrate that the inability of the bNAbs to inhibit capture of primary viruses is independent of the sequence and the stage of infection when the virus was isolated (acute versus chronic) but instead, represents a general characteristic of primary HIV-1.

### 3.2. Capture and Replication of Infectious Molecular Clones in Human PBMCs

The ability of these bNAbs to inhibit IMC virus capture and replication in PBMCs was examined. While 0.1 ng of p24 of primary BaL was sufficient to obtain ~ 50,000 viral RNA copies per million cells after the 1-h incubation and trypsin treatment, the amount of IMC required to obtain the same number of viral copies as primary BaL was 50-fold higher (5 ng p24). Although, a higher amount of IMC was used, and it is important to note that the amount of bNAb present in the virus capture assay was in great excess compared to the number of trimers present on the virus. In contrast to what was observed with the BaL primary virus (Figure 2A), VRC01 inhibited BaL IMC capture by PBMCs (Figure 3A). The percentage of inhibition ranged from 15–50% (Figure 3A), which was significantly higher than the inhibition (1.4–22%) observed with the cognate primary virus (Figure 2A). No significant inhibition of BaL IMC capture was observed with PG9. HIV-1 BaL IMC replication was inhibited by VRC01 (Figure 3B), albeit to varying degrees (45–80%), depending on the donor used, while no inhibition was observed with PG9. Although, somewhat surprising, the inability of PG9 to neutralize BaL in a pseudovirus neutralization assay has been previously reported [16].

To confirm that the inhibition of capture by VRC01 bNAb was applicable to other IMCs also and not restricted to BaL IMC, we utilized acute IMCs, namely C6980V0C22 and 40061.LucR, belonging to subtypes C and CRF01_AE, respectively. C6980V0C22 was originally isolated from an infected person in the acute phase (Fiebig stage I/II) from East Africa [11,17], while 40061 was isolated from the RV217 ECHO study [15]. Compared to Synagis, both PG9 and VRC01 significantly inhibited virus capture of C6980V0C22 by PBMCs from 30–60% (Figure 4A), as well as up to 90% of virus replication (Figure 4B). Similar results were also observed with 40061.LucR with virus capture inhibition ranging from 25–65% (Figure 4C). The bNAbs also inhibited virus replication of 40061.LucR ranging from 50–90% (Figure 4D). Interestingly, the three different IMCs examined appeared to be more susceptible to the CD4-binding site bNAb, VRC01 compared to the quaternary bNAb PG9, which binds to an epitope on the V2 region of the viral envelope.

### 3.3. Virus Capture and Replication of IMCs with Isolated CD4^+^ T Cells

The experiments described above were conducted with bulk PBMCs, which contained CD4^+^ T cells (~65%), CD8^+^ T cells (~6%), B cells (~11%), monocytes (<1%), and NK cells (<1%). Although majority of the bulk PBMCs were HIV-permissive CD4^+^ T cells, we isolated the CD4^+^ T cells and examined virus capture and replication of the IMCs in the absence and presence of bNAbs. Both bNAbs inhibited the capture of C6980V0C22 IMC (Figure 5A) and 40061.LucR IMC (Figure 5C). The inhibition of virus capture by PG9 with C6980V0C22 ranged from 52–55%, with the exception of CD4^+^ T cells from donor 040 where no inhibition of virus capture was seen with PG9 (Figure 5A). VRC01 was slightly more effective at inhibiting virus capture (58–69%) and inhibition was seen with CD4^+^ T cells from all 3 donors (Figure 5A). Both PG9 and VRC01 bNAbs inhibited the replication of C6980V0C22 IMC in all 3 donors with inhibition ranging from 82–89% for PG9 and 93–97% for VRC01 (Figure 5B). Similarly, the two bNAbs inhibited virus capture of 40061.LucR IMC by CD4^+^ T cells from all 3 donors, with inhibition ranging from 29–64% for PG9 and 29–69% for VRC01 (Figure 5C). Both PG9 and VRC01 were also effective in inhibiting virus replication of 40061.LucR IMC, ranging from 77–94% and 69–94%, respectively (Figure 5D). The inhibition observed with the CD4^+^ T cells indicated that the presence of the other immunological cells in the PBMCs did not impact the bNAb’s ability to inhibit virus capture.

## 4. Discussion

Several viruses, including HIV-1, exhibit various patterns of interactions with the cell surface molecules that then lead to virus entry into cells, such as initial weak or nonspecific interactions (surfing) followed by specific attachment to receptors that facilitate local confinement and endocytic uptake of the virus [18,19,20,21,22,23]. This process can be aided by various adhesion proteins, including heparin sulphate proteoglycans and integrins [24] prior to the engagement of specific receptors. In the case of HIV-1, cell-surface membrane fusion has been widely considered as the main route for HIV entry [25,26], although fusion may also occur later, after bound virions have been internalized into endocytic vesicles [27,28,29,30,31,32,33]. Both mechanisms lead to release of the viral capsid into the cytoplasm of the target cell, initiating viral replication and subsequent cellular infection. The typical time scale of the virus entry events from engagement to entry is in the order of minutes [12].

In the present study, we examined the effect of bNAbs on HIV-1 capture and replication using human PBMCs and purified CD4^+^ T cells as target cells to understand the early steps of infection. We found that, unlike the T cell line (A3R5.7) used in the previous study [12], the PBMCS were more permissive to HIV-1 infection. A 10-fold less virus inoculum (p24 concentration) was required to obtain approximately 20,000–50,000 viral RNA copies per million PBMCs. At 48 h post-infection, there was at least a 10-fold increase in the viral RNA copy number with the PBMCs, compared to our previous study with A3R5.7 cells [12] where the increase in viral RNA copies was approximately 3.5-fold.

Several key points emerged from this study: (1) we observed that PBMCs and CD4^+^ T cells captured the primary virus rapidly and the virus was converted to a trypsin-resistant form within minutes post-exposure; (2) the bNAbs inhibited virus replication, but did not inhibit the initial interactions between the primary virus and PBMCs; (3) the inability of bNAbs to inhibit HIV-1 capture was observed with both chronic and acute primary viruses belonging to different subtypes; and (4) there was a dramatic difference in the ability of bNAbs to inhibit IMC capture when compared to primary virus capture, and this was not restricted to the subtype or whether the IMC was derived from a chronic or an acute virus.

The inability of bNAbs to inhibit primary HIV-1 capture is consistent with results observed in *in vivo* challenge experiments with nonhuman primates (NHPs), SHIV-SF162P3, and PGT121, a bNAb targeting the V3-region of HIV-1 env. The bNAb’s ability to inhibit primary virus replication was recapitulated in the NHP experiments by the lack of viremia in the blood. However, the presence of virus was observed in various organs within days of infection, indicating the inability of the bNAb to inhibit entry into target cells in certain organs [34].

The differences in phenotype for the initial interactions of primary HIV-1 and IMCs with the host cell may be attributed to the differences in the structure and composition of proteins and other molecular components present on the virus particle. Primary HIV-1 is produced in PBMCs after multiple rounds of infection and thus consists of a pool of heterogenous quasispecies, whereas IMCs are produced by a single round of virus assembly using a DNA plasmid and transfection of 293T cells, which would result in a relatively homogenous population. Another factor that may influence the initial interactions between the host cell and the virus particle is the difference in the glycosylation patterns present on the HIV-1 envelope trimers produced by the two different cell systems. Recent work by Lu, et al. [35] using single molecule FRET analysis demonstrated the existence of different conformations of HIV-1 envelope trimer. The envelope in the presence of a bNAb was present in conformational state 1, whereas the presence of non-neutralizing antibodies induced a state 3 conformation of the HIV-1 env [35]. Thus, it is possible that the envelope trimers of an IMC may exist in a closed state (state 1), which is CD4 dependent [36], whereas those in a primary virus may exist in multiple states consisting of both open and closed conformations.

Our virus capture data in the presence of VRC01 further demonstrates that the virus envelopes of primary viruses and IMCs interact with the host cell differently. VRC01 targets the CD4 binding site of the envelope. Strong inhibition of the IMC virus capture in the presence of the VRC01 is consistent with the well-established role of CD4 as the primary receptor and the initial point of contact between the IMC and the host cell. However, inability of VRC01 to inhibit primary virus capture on PBMCs and CD4^+^ T cells raises the question of whether CD4 is the initial molecule on the host cell surface that HIV1 interacts with. Another possible explanation for the observed difference between primary viruses and IMCs may be the presence of host proteins on the surface of the primary virions due to budding from the host cell. These host proteins may be responsible for the initial interactions between the host cell and the virus. The results from this study therefore calls for a closer examination of the virus particle, perhaps a detailed proteomics analysis of primary virions and IMCs.

Our results also highlight the importance of reservoir establishment as a concern while using bNAbs to prevent HIV-1 transmission. Although the bNAbs are able to inhibit virus replication, they are unable to prevent the presence of the virus in target cells. Although, only a very minute fraction of these captured viruses is replication competent [12], even a single replication competent virus particle is sufficient for outgrowth and thus could be a cause for concern. Our finding that bNAbs, even a cocktail of multiple bNAbs [12], could not completely block primary virus capture is consistent with a large amount of data reported on reservoir establishment. We speculate that an effective HIV vaccine must elicit virus capture-blocking antibodies as well as bNAbs in order to effectively block HIV-1 acquisition and replication by 100% and thus reduce the chances of a possible reservoir establishment. Therefore, the potential differences in the compositions of the primary viruses and IMCs and their role in initial interactions with cell surface molecules that lead to capture by host cells and virus entry are important considerations in the design of an effective prophylactic vaccine.

## 5. Conclusions

In this work, we demonstrate fundamental differences in the ability of bNAbs to inhibit the capture and replication of primary HIV-1 and IMCs by human PBMCs from healthy donors using a quantitative, kinetically controlled, and high-throughput qRT-PCR assay. While both capture and replication of IMCs were significantly inhibited by bNAbs, in the case of primary HIV-1, replication and not capture was inhibited. Our data indicate that primary HIV-1 and IMCs exhibit distinct phenotypes with respect to their initial interactions with the host cells. We believe that the effectiveness of bNAbs would be further enhanced and 100% inhibition could be achieved with multiple primary HIV-1 if the bNAbs inhibited both capture and replication of primary HIV-1.

## Figures and Tables

**Figure 1 vaccines-08-00782-f001:**
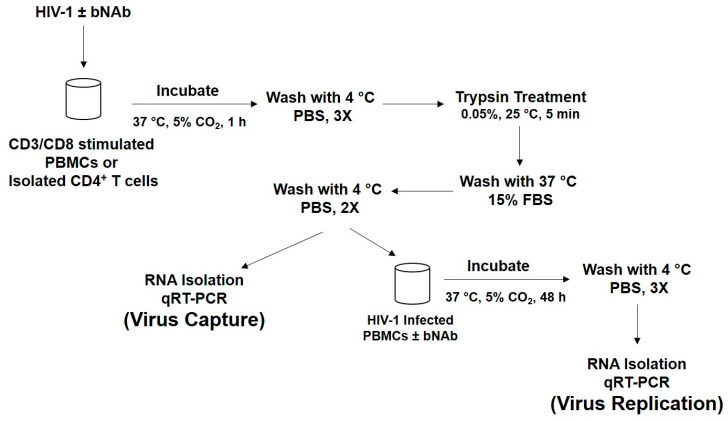
Scheme of the virus capture and replication assay. CD3/CD8 stimulated PBMCs or isolated CD4^+^ T cells were infected with virus (or virus pre-incubated with bNAb) for 1 h at 37 °C/5% CO_2_. The cells were washed three times with cold PBS, and then treated with 0.05% trypsin at room temperature for 5 min. The cells were washed three times and the remaining pellet was used for RNA isolation and qRT-PCR analysis. Cells that were used to examine virus replication were resuspended in media after the trypsin treatment, washed, and incubated for 48 h at 37 °C/5% CO_2_. The cells were then washed three times with cold PBS and then processed for RNA isolation and qRT-PCR analysis.

**Figure 2 vaccines-08-00782-f002:**
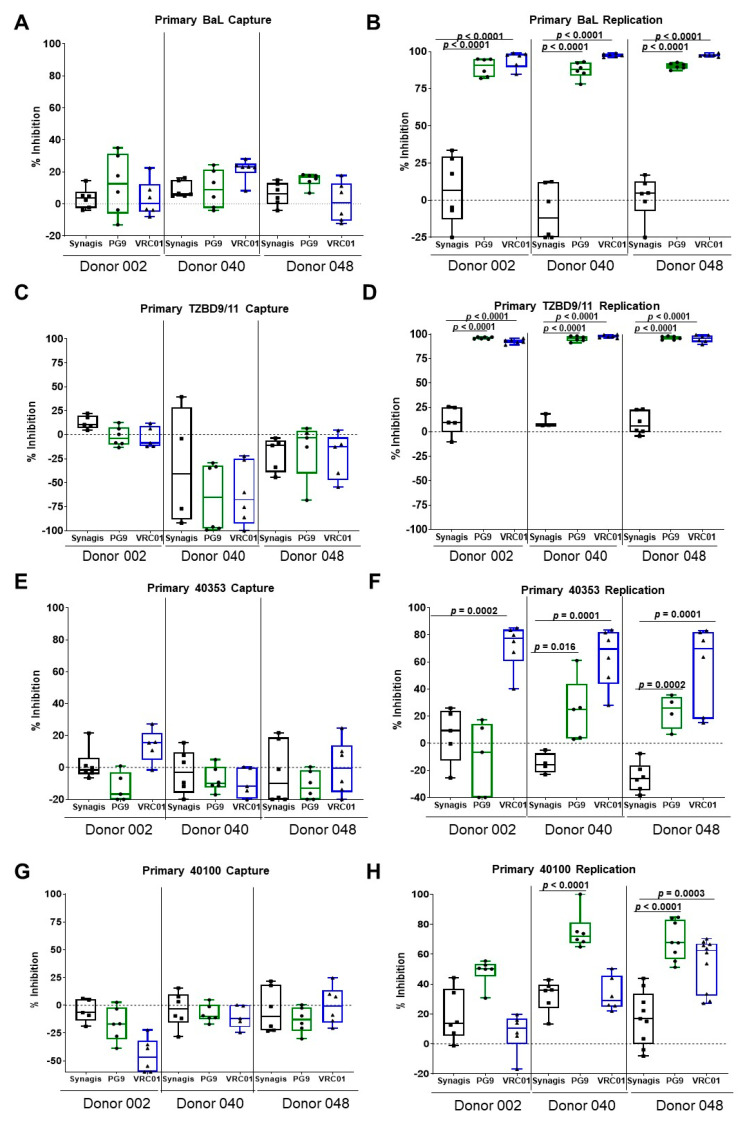
bNAbs do not inhibit virus capture of primary HIV-1 BaL, TZBD9/11, acute 40353, and acute 40100 but do inhibit virus replication in PBMCs. (**A**) PG9 and VRC01 did not inhibit the capture of primary subtype B, BaL, to the three donor PBMCs examined. (**B**) The bNAbs inhibited the replication of primary BaL after 48 h of infection for all three donor PBMCs. (**C**) Virus capture by the subtype C primary virus, TZBD9/11, was not inhibited by either PG9 or VRC01. (**D**) Both bNAbs, PG9 and VRC01 significantly inhibited (*p* < 0.0001) virus replication of TZBD9/11 in all three donor PBMCs examined. (**E**) The two bNAbs were unable to inhibit virus capture of the subtype B acute primary virus 40353, from RV217 ECHO study, with all three donor PBMCs examined. (**F**) PG9 was able to significantly inhibit virus replication of 40353 in two out of the three donor PBMCs examined. VRC01 was more effective compared to PG9 and significantly inhibited virus replication of 40353 in all three donor PBMCs examined. (**G**) Virus capture by the subtype CRF01_AE acute primary virus 40100, from the RV217 ECHO study, was not inhibited by either PG9 or VRC01. (**H**) PG9 significantly inhibited (*p* < 0.0001) virus replication of 40100 in two out of three donor PBMCs examined. VRC01 was able to significantly inhibit (*p* = 0.0003) virus replication of 40100 with PBMCs only from donor 048. The data are shown as box-whisker plots (the box extends from the 25th to 75th percentiles; the whiskers contain the min to max showing all points). The data shown are from at least 2 independent experiments performed in triplicate.

**Figure 3 vaccines-08-00782-f003:**
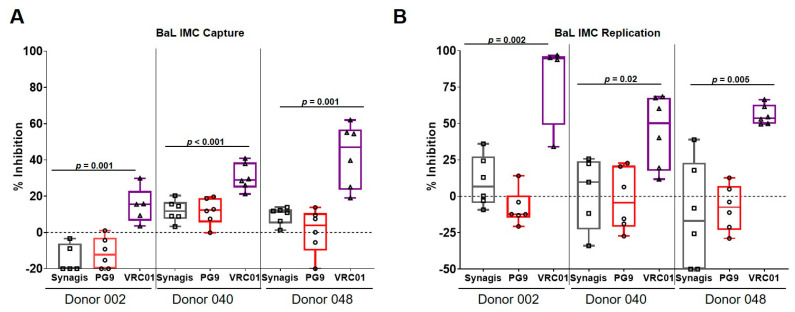
bNAbs inhibited virus capture and replication of BaL IMC. (**A**) Unlike its primary virus counterpart, VRC01 significantly (*p* = 0.001) inhibited the capture of BaL IMC in the three donor PBMCs examined, albeit to varying degrees compared to the negative control monoclonal antibody, Synagis. (**B**) VRC01 inhibited BaL IMC replication in the three different donor PBMCs. However, no inhibition was observed with PG9. The data are shown as box-whisker plots (the box extends from the 25th to 75th percentiles; min to max showing all points). The data shown are from at least 2 independent experiments performed in triplicate.

**Figure 4 vaccines-08-00782-f004:**
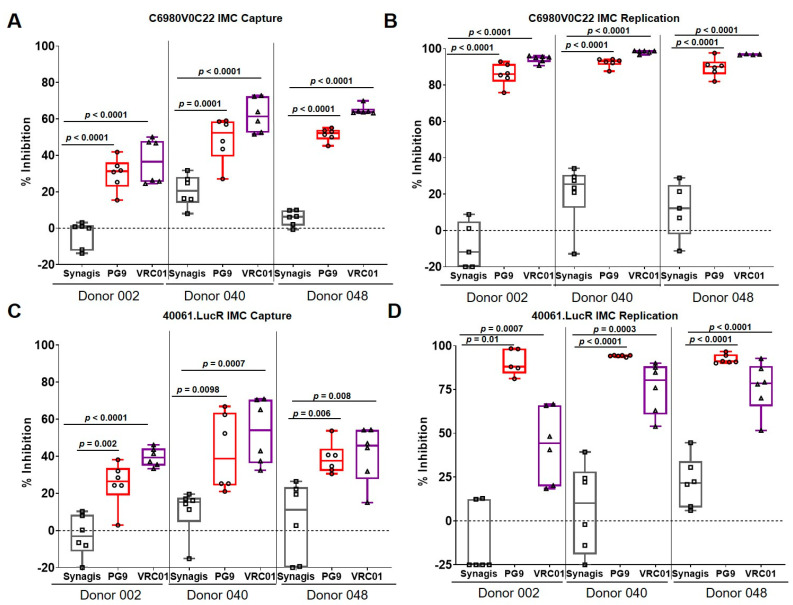
bNAbs inhibited virus capture and replication of C6980V0C22 and 40061.LucR IMCs. (**A**) PG9 and VRC01 significantly (*p* < 0.001) inhibited virus capture of the subtype C IMC, C6980V0C22 in all three donor PBMCs. (**B**) The two bNAbs also significantly (*p* < 0.001) inhibited virus replication of all three donors examined. (**C**) PG9 and VRC01 significantly (*p* < 0.01) virus capture of the CRF01_AE IMC, 40061.LucR for all three donor PBMCs. (**D**) The bNAbs also significantly inhibited virus replication of the IMC in the bulk PBMCs examined. The data are shown as box-whisker plots (the box extends from the 25th to 75th percentiles; min to max showing all points). The data shown are from at least 2 independent experiments performed in triplicate.

**Figure 5 vaccines-08-00782-f005:**
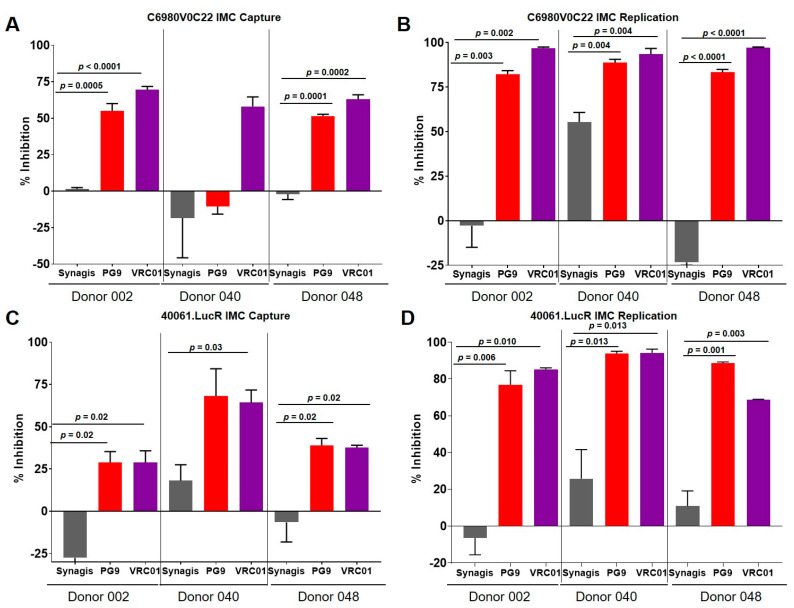
bNAbs inhibited virus capture and replication of acute IMCs in isolated CD4^+^ T cells. (**A**) PG9 inhibited virus capture of the subtype C IMC, C6980V0C22, in purified CD4^+^ T cells in two out of the three donors examined. VRC01 inhibited virus capture of C6980V0C22 in CD4^+^ T cells from all three donors. (**B**) The two bNAbs significantly inhibited virus replication of C6980V0C22 in CD4^+^ T cells from the three donors. (**C**) The two bNAbs inhibited capture of the subtype CRF01_AE IMC, 40061.LucR with the CD4^+^ T cells from all three donors. (**D**) Both PG9 and VRC01 inhibited virus replication of 40061.LucR and CD4^+^ T cells. Significant inhibition (*p* = 0.01 or *p* < 0.01) was observed for both bNAbs and with all three donors examined. The data are shown as bar graphs representing mean ± SEM and are from at least 2 independent experiments performed in triplicate.

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
