# Peer review of "Primary HIV-1 and Infectious Molecular Clones Are Differentially Susceptible to Broadly Neutralizing Antibodies"

_vaccines, 2020, doi:10.3390/vaccines8040782_

Round 1

Reviewer 1 Report

This manuscript by Kim et al. builds on previous work by the authors who developed a qRT-PCR assay allowing to study the ability of bNAbs to prevent HIV entry into T cells very early following inoculation. Here, they use PBMC and purified CD4+ T cells instead of a T cell line to confirm the physiological relevance of their previous observations indicating that primary HIV strains are not as susceptible as IMC to entry blockade by bNAbs. This manuscript is well-written with thoroughly detailed methods and reports interesting results of importance for the development of HIV prevention strategies based on bNAbs. My specific comments and suggestions can be found below:

Major comments:

1) Results section 3.1.: I would suggest removing some of the technical details from the first paragraph as the assay is thoroughly described in Materials and Methods and in Figure 1.

2) Fig. 2: The graph displayed in 2A is the only graph not displaying negative values. Were negative values adjusted to 0 for this graph? If that is the case, 0 values should be represented as negative values to match the other graphs and facilitate comparison.

3) Lines 204-206: I would suggest rephrasing this sentence. Do the authors mean “the ability of bNAbs to inhibit capture of primary acute HIV-1”?  

4) I would suggest combining Fig. 2 and 3 into one.

5) Fig. 3B/Lines 218-220: with PBMCs from donor 040, inhibition of replication appears comparable between Synagis and VRC01. It might be more accurate to state that VRC01 significantly inhibited replication in 1 out of 3 donor PBMCs.

6) It would have been interesting to test if bNAbs have enhanced ability to block capture/replication of BaL IMC or primary strains when using isolated CD4+ T cells as targets. Did the authors look into that?

7) Since one of the IMC is of subtype C, it would have been interesting to also include analysis of a primary HIV subtype C in this study.

Minor comments:

1) For consistency, the subtypes of all IMCs should be mentioned in Materials and Methods.

2) Are the values for the percentages of inhibition mentioned throughout the text (i.e., line 240) the median percentages of inhibition? This should be indicated.

3) It appears that bNAbs have an overall decreased ability to inhibit primary HIV replication when using PBMC from donor 002. Is there any difference in the phenotype of CD4+ T cells from this donor or any other factor in this donor that may explain a potential enhanced permissivity to primary HIV?

Author Response

Response to Reviewer 1

Reviewer #1

This manuscript by Kim et al. builds on previous work by the authors who developed a qRT-PCR assay allowing to study the ability of bNAbs to prevent HIV entry into T cells very early following inoculation. Here, they use PBMC and purified CD4+ T cells instead of a T cell line to confirm the physiological relevance of their previous observations indicating that primary HIV strains are not as susceptible as IMC to entry blockade by bNAbs. This manuscript is well-written with thoroughly detailed methods and reports interesting results of importance for the development of HIV prevention strategies based on bNAbs. My specific comments and suggestions can be found below:

Response: We thank the reviewer for the positive comments.

Major comments:

1) Results section 3.1.: I would suggest removing some of the technical details from the first paragraph as the assay is thoroughly described in Materials and Methods and in Figure 1.

Response to Major Comment 1: We have removed some of the technical details of the assay from the first paragraph as suggested by the reviewer and condensed it (lines 177-179) in the revised manuscript.

2) Fig. 2: The graph displayed in 2A is the only graph not displaying negative values. Were negative values adjusted to 0 for this graph? If that is the case, 0 values should be represented as negative values to match the other graphs and facilitate comparison.

Response to Major Comment 2: We thank the Reviewer for catching this error in Figure 2A. There are negative values in this graph, and therefore we have changed it to include all data points and have revised the figure.

3) Lines 204-206: I would suggest rephrasing this sentence. Do the authors mean “the ability of bNAbs to inhibit capture of primary acute HIV-1”?

Response to Major Comment 3: We do mean “the ability of bNAbs to inhibit capture of primary acute HIV-1”, therefore we have changed the sentence to more accurately reflect this statement (lines 220-222) in the revised manuscript.  We have rephrased the sentence to read as “To confirm that the inability of bNAbs to inhibit primary HIV-1 BaL (a chronic virus) capture was not due to the time of isolation based on the infection status (chronic), we examined the ability of bNAbs to inhibit virus capture of primary acute HIV-1”.

4) I would suggest combining Fig. 2 and 3 into one.

Response to Major Comment 4: As the Reviewer has suggested, we have combined Figures 2 and 3. We have changed the Figure Legend to incorporate this change, as well as made subsequent changes to the numbering of the Figures following Figure 2 in the revised manuscript.

5) Fig. 3B/Lines 218-220: with PBMCs from donor 040, inhibition of replication appears comparable between Synagis and VRC01. It might be more accurate to state that VRC01 significantly inhibited replication in 1 out of 3 donor PBMCs.

Response to Major Comment 5: We agree with the reviewer and have made the change as suggested (Line 236-237). The sentence now reads as follows: “PG9 inhibited replication of the CRF01_AE virus 40100 for all three donor PBMCs (48-75%), while VRC01 inhibited replication in one out of three donor PBMCs (63%) (Figure 2H).”

6) It would have been interesting to test if bNAbs have enhanced ability to block capture/replication of BaL IMC or primary strains when using isolated CD4+ T cells as targets. Did the authors look into that?

Response to Major Comment 6: We thank the reviewer for this question. We are currently planning to do these experiments this as part of a larger study.

7) Since one of the IMC is of subtype C, it would have been interesting to also include analysis of a primary HIV subtype C in this study.

Response to Major Comment 7: We thank the Reviewer for this comment, and we had this data but did not include it in the original manuscript as it was not different from the other primary HIV-1. We have now included the capture and replication data for primary TZBD9/11 (subtype C) in Figures 2C and 2D.

Minor comments:

1) For consistency, the subtypes of all IMCs should be mentioned in Materials and Methods.

Response to Minor Comment 1: As suggested by the Reviewer, we have added the subtypes of the IMCs in the Materials and Methods section (lines 94 and 98).

2) Are the values for the percentages of inhibition mentioned throughout the text (i.e., line 240) the median percentages of inhibition? This should be indicated.

Response to Minor Comment 2: For Figures 2, 3, and 4, the box whisker plots depict the median values for the percentage of inhibition and where these are mentioned in the text, they represent the median values. However, Figure 5 shows the mean percentages of inhibition as bar graphs. Therefore to clarify this, we have added a statement, “Reported percentage inhibition values are the median values in the box whisker plots or mean ± SEM in the bar graphs,” in the Statistical Analysis (line 166-167) of the revised manuscript.

3) It appears that bNAbs have an overall decreased ability to inhibit primary HIV replication when using PBMC from donor 002. Is there any difference in the phenotype of CD4+ T cells from this donor or any other factor in this donor that may explain a potential enhanced permissivity to primary HIV?  

Response to Minor Comment 3: While this is an interesting question, at the present time we do not have an explanation for the decreased ability of donor 002 PBMCs to inhibit primary HIV-1 infection by bNAbs. We agree that this should be explored further as part of a separate study.

Reviewer 2 Report

The authors show, with the use of qPCR based quantification of viral RNA, that two bNABs PG9 and VRC01 do not prevent capture of HIV, however replication of the virus is affected. Presented data clearly supports those claims. Moreover, differences in virus capture depending on the cellular origin (transfected 293Ts, primary isolates etc.) are shown to clearly affect capture of the virus.

The authors dispute validity of standard TZM-bl assay used for infectivity and propose that virus capture assay can provide additional insight into early events of virus-host cell interactions. As valid as those claims are, the authors do not show direct comparison of the methods while indeed they are meant to measure two different things.

HIV can be endocytosed and find itself within trypsin-resistant compartment. That alone could explain why the authors see the presence of HIV inside the cells following bNAb binding. An elegant nd simple control experiment to rule out that possibility would be to incubate the virus with the bNAbs at 4°C in addition to the 37°C used. Moreover, envelope deficient HIV could be used to verify if RNA can be detected independent of bNAb binding.

Importantly, virus capture does not equal infection. It is likely that the replication block reflects the blockage at the level of infection.

As such, the paper does indeed show that virus capture can occur in presence of bNAbs, but claims that such antibody-opsonized virus can establish a reservoir are not supported by experimental evidence (such as comparison of intergrated proviral DNA between the Synagis antibody and PG9 or VRC0).

Author Response

The authors show, with the use of qPCR based quantification of viral RNA, that two bNABs PG9 and VRC01 do not prevent capture of HIV, however replication of the virus is affected. Presented data clearly supports those claims. Moreover, differences in virus capture depending on the cellular origin (transfected 293Ts, primary isolates etc.) are shown to clearly affect capture of the virus.

Response to Reviewer: We thank the reviewer for the positive comments.

The authors dispute validity of standard TZM-bl assay used for infectivity and propose that virus capture assay can provide additional insight into early events of virus-host cell interactions. As valid as those claims are, the authors do not show direct comparison of the methods while indeed they are meant to measure two different things.

Response to Reviewer: We would like to clarify our statement regarding the standard TZM-bl assay. In fact, our results with regards to the virus replication data from our assay are in agreement with previous studies using the TZM-bl assay. The TZM-bl assay is a HeLa cell line assay that utilizes pseudoviruses. The cell line has been engineered to overexpress CD4 and co-receptors for HIV-1 infection. This could allow for the misinterpretation of the mechanism of inhibition of bNAbs. We want to highlight the importance of utilizing primary HIV-1 because that clearly distinguishes virus capture and entry.

HIV can be endocytosed and find itself within trypsin-resistant compartment. That alone could explain why the authors see the presence of HIV inside the cells following bNAb binding. An elegant nd simple control experiment to rule out that possibility would be to incubate the virus with the bNAbs at 4°C in addition to the 37°C used. Moreover, envelope deficient HIV could be used to verify if RNA can be detected independent of bNAb binding.

Response to Reviewer: We agree with the reviewer and we also hypothesize that the virus is endocytosed into a trypsin resistant compartment. We propose to examine this in great detail in our future studies, but it is outside the scope of the present study. In our previous publication (doi:10.1016/j.virol.2017.05.015) using A3R5 cells (a T cell line that expresses CD4, CXCR4, CCR5, and the integrin homing receptor α4β7, in copy numbers similar to that observed on human PBMCs), we demonstrated that capture was envelope dependent since neither free RNA nor the virus particles lacking the envelope showed significant capture. We believe the same will also be true for PBMCs.

Importantly, virus capture does not equal infection. It is likely that the replication block reflects the blockage at the level of infection.

Response to Reviewer: We agree with the reviewer that virus capture does not equal infection. However, we do believe that a percentage of virus particles that have been captured and internalized into a trypsin resistant state is infectious and can reflect viral replication and infection.

As such, the paper does indeed show that virus capture can occur in presence of bNAbs, but claims that such antibody-opsonized virus can establish a reservoir are not supported by experimental evidence (such as comparison of intergrated proviral DNA between the Synagis antibody and PG9 or VRC0).

Response to Reviewer: We appreciate the reviewer’s comments, and we suggest that perhaps a portion of the virus that has entered the cells may have the potential to establish a reservoir. However, we agree that further investigation is warranted regarding the presence of integrated proviral DNA in future studies. A single replication competent virion is sufficient to be the seed of a reservoir and unless a bNAb can efficiently block capture and replication 100%, the possibility exists for the potential establishment of a reservoir.   

Reviewer 3 Report

Couple real small things but this is a high-quality paper for sure:

41: comma after neutralization

43: days post infection

52: drop “in the present study”

54: and/or

321: “trypsin-resistant state” feels off

Great graphs

Author Response

Response to Reviewer #3

Couple real small things but this is a high-quality paper for sure:

Response to Reviewer: We thank the reviewer for their positive remarks regarding our manuscript. We have made the changes as requested and highlighted the changes in the Word document for the following comments:

41: comma after neutralization

Response to Reviewer: We have inserted a common after neutralization.

43: days post infection

Response to Reviewer: We have changed “days after infection” to days post infection as suggested.

52: drop “in the present study”

Response to Reviewer: We have deleted “in the present study”

54: and/or

Response to Reviewer: Inserted / between and or

321: “trypsin-resistant state” feels off

Response to Reviewer: We have changed the sentence to “the virus was converted to a trypsin-resistant  form” This is a term that was have used in our previous publication (doi:10.1016/j.virol.2017.05.015) to accurately describe the virus particles that are present inside the cell, and therefore are trypsin-resistant.

Round 2

Reviewer 2 Report

I thank the authors for replies. We differ in our opinion on validity of TZM-bl assay (which can be used also with primary isolates), but we agree that the virus can be found inside the cells following opsonisation by bNABs. Whether this could contribute to reservoir establishment is at this stage a speculation and remains to be verified experimentally

I will be looking forward to follow up experiments. Good luck!
